# Noradrenergic Components of Locomotor Recovery Induced by Intraspinal Grafting of the Embryonic Brainstem in Adult Paraplegic Rats

**DOI:** 10.3390/ijms21155520

**Published:** 2020-08-01

**Authors:** Anna Kwaśniewska, Krzysztof Miazga, Henryk Majczyński, Larry M. Jordan, Małgorzata Zawadzka, Urszula Sławińska

**Affiliations:** 1Nencki Institute of Experimental Biology, Polish Academy of Sciences, 02-093 Warsaw, Poland; a.kwasniewska@nencki.edu.pl (A.K.); k.miazga@nencki.edu.pl (K.M.); h.majczynski@nencki.edu.pl (H.M.); m.zawadzka@nencki.edu.pl (M.Z.); 2Department of Physiology and Pathophysiology, University of Manitoba, Winnipeg, MB R3E 0J9, Canada; Larry.Jordan@umanitoba.ca

**Keywords:** spinal rats, transplantation, embryonic brainstem, serotonin, α_2_ adrenergic receptors

## Abstract

Intraspinal grafting of serotonergic (5-HT) neurons was shown to restore plantar stepping in paraplegic rats. Here we asked whether neurons of other phenotypes contribute to the recovery. The experiments were performed on adult rats after spinal cord total transection. Grafts were injected into the sub-lesional spinal cord. Two months later, locomotor performance was tested with electromyographic recordings from hindlimb muscles. The role of noradrenergic (NA) innervation was investigated during locomotor performance of spinal grafted and non-grafted rats using intraperitoneal application of α_2_ adrenergic receptor agonist (clonidine) or antagonist (yohimbine). Morphological analysis of the host spinal cords demonstrated the presence of tyrosine hydroxylase positive (NA) neurons in addition to 5-HT neurons. 5-HT fibers innervated caudal spinal cord areas in the dorsal and ventral horns, central canal, and intermediolateral zone, while the NA fiber distribution was limited to the central canal and intermediolateral zone. 5-HT and NA neurons were surrounded by each other’s axons. Locomotor abilities of the spinal grafted rats, but not in control spinal rats, were facilitated by yohimbine and suppressed by clonidine. Thus, noradrenergic innervation, in addition to 5-HT innervation, plays a potent role in hindlimb movement enhanced by intraspinal grafting of brainstem embryonic tissue in paraplegic rats.

## 1. Introduction

It is well known that monoaminergic systems (noradrenergic and serotonergic) play crucial roles in mammalian locomotor activity. There has been considerable evidence that the noradrenergic system plays an important role in locomotion in cats. For example, Jankowska with co-workers [1,2] showed that in spinalized and paralyzed cats, L-Dopa could induce fictive locomotion. The first demonstration that the activation of α_2_ noradrenergic receptors by clonidine can induce locomotor activity in spinal cats was published by Forssberg and Grillner in 1973 [3]. In cats, following complete thoracic spinal cord transection, both intraperitoneal [4], and intrathecal [5] application of clonidine initiated treadmill locomotion shortly after spinalization (as early as the second postoperative day). Clonidine also improved locomotor training in adult spinal cats and considerably accelerated the recovery of hindlimb locomotion [6]. However, the effects of clonidine application had opposite effects in cats with partial lesions [7], where locomotor abilities were significantly deteriorated.

Grafting of brainstem embryonic cells, serotonergic (5-HT) from caudal raphe nuclei or noradrenergic (NA) from locus coeruleus, into the spinal cord of adult spinal rats significantly enhances the recovery of hindlimb movements. It was demonstrated that this recovery is associated with re-established serotonergic [8,9,10,11,12] or noradrenergic [13,14] innervation of the spinal cord below the lesion. The role of NA neurons in such a graft was suggested after grafting pure 5-HT neuron suspension when NA neurons were killed using 6-hydroxydopamine, resulting in less effective locomotor recovery than that produced by a grafted suspension of 5-HT/NA neurons without the toxin [15]. The receptor mechanisms involved in 5-HT related locomotor recovery were investigated using various agonists (8-hydroxy-2-(di-*N*-propylamino)-tetralin; quipazine) in spinal rats [16,17,18,19] and antagonists (Cyproheptadine; SB 269970) in spinal grafted rats [11,12]. There is less information about noradrenergic mechanisms involved in the control of locomotor functions in grafted rats.

These two cell groups are in close proximity in the portion of the medulla [20,21] that we use for the grafts, and so there is likely to be a population of NA neurons in the graft even when we attempt to maximize the 5-HT cell content. Here we attempt to examine the contribution of any NA neurons in our grafts using the α_2_ adrenergic receptor agonist clonidine and the antagonist yohimbine. We strove to determine if a combination of 5-HT and NA neurons in the graft offered any advantage to the recovery of locomotion.

Our results indicate a potent role of the NA innervation in locomotor recovery in paraplegic rats with the intraspinal graft of embryonic brainstem tissue. In addition to the important role of 5-HT neurons in this process, our findings provide new insights into the noradrenergic mechanisms underlying the locomotor recovery in rats. A preliminary report of these findings has been published in an abstract form [22].

## 2. Results

### 2.1. 5-HT and NA Neurons in the Donor Brainstem

We analyzed the brainstem area of the E14 embryo (when the day following mating is designed as embryonic day zero, E0) that was dissected for the grafting procedure. We estimated that the region of interest contains typically 110,000–120,000 cells of which approximately 3% are serotonin (5-HT) neurons medially located in the B1/B2/B3 groups. In close lateral proximity there are NA neurons in the AC1/AC2/AC3 groups [20,21]. This suggested that in the piece of brainstem dissected for grafting we harvested the NA neurons in addition to 5-HT neurons with high probability. An example of the embryonic brainstem with the approximate region of the tissue taken for grafting marked with the green quadrangle is shown in Figure 1.

### 2.2. 5-HT and NA Neurons in the Grafted Spinal Cord

First, we investigated the graft tissue survival and its possible integration with the host spinal cord environment. In all spinal transplanted rats, the presence of grafted tissue was positively verified. The grafted tissue was detected predominantly in the gray matter of the spinal cord as aggregates with about 1000–1500 µm in diameter surrounded by host tissue (Figure 2). Detection of the robust serotonergic fibers infiltrating the host spinal cord areas outside the graft region (i.e., the ventral horns, Figure 2A) indicates successful integration of the grafted tissue within the host spinal cord environment. On the other hand, GFAP (glial fibrillary acidic protein) staining demonstrated the absence of reactive astrocytes on the border of the grafted tissue and the host spinal cord (Figure 2B). These results indicate that embryonic tissue easily integrates within host spinal cord and does not evoke long-lasting astrogliosis that, if present, might have created a solid scar. 

Next, we investigated the presence of NA and 5-HT neurons in the spinal cord of grafted spinal rats at 3 months after the grafting procedure. The presence of fetal brainstem grafts was confirmed in all grafted rats with observed substantial hindlimb locomotor recovery (for details of behavioral and functional investigation see the following sections). Each graft contained 5-HT/TPH-positive neuronal cell bodies (Figure 3A,B) that gave rise to 5-HT fibers infiltrating the grafts and extending for some distance into the host spinal cord (Figure 3C–I). Detailed examination of serial sections immunostaining revealed that 5-HT/TPH neurons preferentially occupy areas in close proximity to the rim of the grafted tissue. We quantified serotonergic neurons and found from 106 to 464 5-HT/TPH-positive cells per graft with the length ranged between 720 to 1440 µm. On average, we identified 270 ± 143 5-HT/TPH cells per graft, which means that the viability of the fetal cells at 3 months after transplantation accounts for around 10%. However, it needs to be noticed that both total cell number and their viability was underestimated since cells were counted in every fifth section of 12 µm thickness, each 70 µm apart (*n* = 5 grafted rats). The typical targets of 5-HT fibers of graft origin were the areas of the dorsal and ventral horn, central canal, and intermediolateral zone, while the spinal control rats without a graft were devoid of 5-HT innervation completely (not shown in the figure). In the grafted rats the 5-HT fibers were found as much as 20 mm from the graft caudally in the host spinal cord approximately in the S1 segment (Figure 3D–I). Within the mentioned area of grafted spinal cords, we identified from 7 to 164 NA (TH positive) neurons, on average 87 ± 67 cells per graft (Figure 3J, notice that the total number of NA cells was underestimated as in case of 5-HT/TPH-positive cells). The NA fibers have similar targeting in the intact spinal cord as 5-HT fibers [8,23] but in our grafted rats their distribution was weaker and limited mainly to the areas of the central canal and the intermediolateral zone (Figure 3J–O). The most caudal NA innervation in the grafted spinal cord was detected around the central canal with single fiber visibility in the MN area of the ventral horn at about the L3 spinal cord segment (Figure 3M–O).

Using EdU (5-Ethynyl-2-deoxyuridine) injected to the pregnant rat at E13 we found that 5-HT and NA neurons were already postmitotic at E14 when the brainstem tissue was harvested for grafting since we did not observe accumulation of EdU in cells that gave rise to neurons at E14 (Figure 3A,B,J). 

We demonstrated that the 5-HT and NA neurons identified in the graft vicinity were surrounded by each other’s axons (Figure 4A–D). These data suggest that the reciprocal interaction of both types of monoaminergic neurons might be essential for the functional positive effect of the embryonic brainstem intraspinal grafting. To confirm this possibility, we performed pharmacological testing using the adrenergic agonist, clonidine and antagonist, yohimbine, in chronic paraplegic rats with and without the graft.

### 2.3. Adrenergic α_2_ Receptors in Control of Locomotor Plantar Stepping in Spinal Grafted and non-Grafted Rats—General Considerations

Next, we tested the role of a noradrenergic (NA) component in the locomotor recovery of paraplegic rats that have received the intraspinal graft of embryonic brainstem tissue. To do this the locomotor abilities were tested in the grafted and non-grafted rats before and after IP (intraperitoneal) application of either yohimbine, noradrenergic α_2_ antagonist, or clonidine, α_2_ agonist. Locomotor abilities were tested with animals suspended over the treadmill with their feet touching the moving belt with a speed of 10 cm/s. As we described before [10,11,24] during such tests tail pinching induces rhythmic plantar stepping in spinal grafted rats, which is accompanied by the regular EMG burst activity of the soleus (Sol) and tibialis anterior (TA) muscles of both hindlimbs (see Figure 5A,D). 

Administration of yohimbine substantially improved locomotor hindlimb movements in grafted spinal rats by lengthening the episodes of regular, sustained plantar stepping (Figure 5B,C; see also the following analysis of the number of proper plantar steps). The recovered plantar stepping of spinal grafted rats was significantly impaired by clonidine application (Figure 5E,F), resulting in deterioration of body weight support and dragging both hindlimbs over the moving treadmill belt. It is important to note that after clonidine treatment the grafted rats became less sensitive to tail stimulation and the bursting EMG activity was sustained for a brief period of time after starting the tail stimulation (see red squares marking such episodes in Figure 5E,F). 

In control spinal rats without the graft, the application of yohimbine induced slight but not significant alterations in the locomotor abilities, while after clonidine the hindlimbs were significantly less active than before treatment. Hardly any movement was obtained in the first 20–40 min after the drug administration (Figure 5K). Then we observed some recovery of hindlimb dragging over the moving treadmill belt, however still without the presence of any plantar stepping as it was in the pre-drug conditions (Figure 5L). 

To quantify the effect of α_2_ agonist and antagonist on locomotor performance of spinal grafted and non-grafted rats, we established the number of proper plantar steps expressed by animals during the 40 s period with the best hindlimb movement selected out of 1 min locomotor test before and after drug application. A proper plantar step was defined as the left hindlimb step with obvious paw contact with the ground and substantial body weight support, with no toe dragging, and followed by a proper step cycle of the contralateral hindlimb. The analysis was done independently by two investigators (one blind) inspecting the video recording carried out during the experimental procedure. Comparison of yohimbine vs. clonidine effects in spinal grafted rats confirmed an increased number of proper plantar steps (Figure 6A) performed during the 40 s locomotor test in a short time (< 30 min) after yohimbine (42.33 ± 3.45 vs. 52.67 ± 2.76; *RM ANOVA* F_(5,25)_ = 127.1, *p* = 0.0011 with *post-hoc Tukey’s multiple comparisons test, p* = 0.02), while clonidine treatment reduced dramatically the number of proper plantar steps (Figure 6B) in a short time (< 30 min) and later (~1 h) after drug application (48.67 ± 3.74 vs. 2.17 ± 0.79 vs. 3.5 ± 1.69 respectively; *post-hoc Tukey’s multiple comparisons test, p* < 0.0001 in both time points). Moreover, the number of proper plantar steps after clonidine treatment in spinal grafted rats was significantly smaller than that obtained in the same rats before as well as after yohimbine treatment (*post-hoc Tukey’s multiple comparisons test, p* < 0.0001 for all experimental conditions). Comparison of effects of either drug applications in spinal non-grafted rats, demonstrated a reduction of the number of proper plantar steps after clonidine applications (Figure 6D), which was significant however only in a short time after drug application (9.8 ± 1.85 vs. 4.8 ± 1.62; F_(5,20)_ = 4.69, *p* = 0.009; *post-hoc Tukey’s multiple comparisons test, p* < 0.05). One hour after clonidine application the reduction from 9.8 ± 1.85 to 5.4 ± 1.33 was not significant (*post-hoc Tukey’s multiple comparisons test p* = 0.09). The yohimbine application did not induce any significant changes in the number of steps (Figure 6C; *post-hoc Tukey’s multiple comparisons test, p* > 0.05). The results obtained in spinal grafted rats are similar to those obtained with these drugs in cats with a spinal partial lesion [7], suggesting that the graft’s ability to restore some innervation lost after spinalization results in a situation resembling a partial lesion.

### 2.4. Adrenergic α_2_ Receptors in Control of Locomotor Plantar Stepping in Spinal Rats—EMG Activity Analysis

EMG recordings from ankle extensor (Sol) and flexor (TA) muscles of both hindlimbs performed in all the experimental animals gave us an opportunity to investigate the locomotor indices: the step cycle duration, burst duration, and interlimb coordination.

#### 2.4.1. Step Cycle and Burst Durations

We found that in spinal grafted rats yohimbine increased slightly the step cycle duration (Figure 7A). *RM ANOVA* (F_(2,22)_ = 7.386, *p* = 0.0035) confirmed statistical significance of step cycle prolongation 610.5 ± 92.16 ms up to 673.2 ± 80.66 ms in a short time after drug application (< 30 min) with *post-hoc Tukey’s multiple comparisons test* (*p* = 0.025) and later (~1 h) after drug application up to 691.2 ± 64.98 ms (*p* = 0.0035). In the grafted rats after clonidine treatment, we obtained significant deterioration of plantar stepping which was expressed at first by limited bursting EMG activity for a brief period of time after the drug took effect and before plantar stepping stopped completely (see red squares marking such episodes in Figure 5E,F). We took these episodes before the hindlimb movement stopped for quantifying the cycle duration and the burst duration in EMG activity. There was a tendency for increased step cycle duration after clonidine application (Figure 7B); however, this increase was not significant (F_(2,22)_ = 2.258, *p* = 0.1282). At the same time, the EMG burst duration of the Sol muscle remained unchanged after yohimbine (Figure 7C,E; F_(5,55)_ = 6.379, *p* < 0.0001; *Tukey’s multiple comparisons test p* > 0.05), while after clonidine in a short time after drug application the Sol EMG burst duration was shortened significantly from 405.0 ± 90.18 ms to 302.6 ± 50.62 ms (Figure 7D; *Tukey’s multiple comparisons test p* < 0.02). In addition, *Tukey’s multiple comparisons* showed that the EMG burst duration of the Sol muscle was significantly longer before and after yohimbine than that after clonidine application (Figure 7C,D; *p* < 0.05; *p* < 0.001; *p* < 0.0001 respectively). The TA EMG burst was prolonged significantly (F_(5,55)_ = 3.988, *p* = 0.0037) after clonidine from 113.5 ± 19.5 ms up to 201.3 ± 141.9 ms (Figure 7F; *Tukey’s multiple comparisons test p* < 0.02) but not after yohimbine (Figure 7E; *Tukey’s multiple comparisons test p* > 0.05). In addition, *Tukey’s multiple comparisons* showed that the EMG burst duration of the TA muscle was significantly longer after clonidine (< 30 min) than that after yohimbine application (Figure 7E,F; *p* < 0.02 at both time points).

Although the spinal non-grafted rats presented only very occasional hindlimb plantar stepping, the EMG activity recorded during hindlimb movement induced by tail pinching show rhythmic activity that could be used to establish the cycle duration (defined as the time duration between the two consecutive TA EMG busts) as well as the EMG burst duration of the Sol and TA muscles. In spinal non-grafted rats yohimbine as well as clonidine treatment prolonged the step cycle significantly (Figure 7G,H; F_(4,32)_ = 6.854, *p* = 0.0004). It is important to note that after clonidine application in non-grafted rats the hindlimb movement was dramatically deteriorated and hardly any movement was observed for more than 40 min after drug application. Thus, we were able to analyze locomotor indices only 1 h after drug application when there was some recovery of hindlimb movements obtained and at least ten consecutive EMG bursts of hindlimb muscle activity was recorded during the locomotor test. Yohimbine application prolonged the step cycle from 443.4 ± 57.02 up to 592.8 ± 133.9 in short time and after 1 h up to 607.8 ± 176.2 (*Tukey’s multiple comparisons test p* < 0.05 for both time points) and from 414.3 ± 69.99 to 614.9 ± 123.2 (*p =* 0.005) 1 h after clonidine. After clonidine treatment (Figure 7J,L), but not after yohimbine (Figure 7I,K), the EMG burst duration was prolonged in both Sol and TA muscles. One hour after treatment the Sol EMG burst was prolonged from 208.7 ± 38.96 ms to 294.6 ± 45.58 ms (F_(4,36)_ = 4.144, *p* = 0.0073; *Tukey’s multiple comparisons test p* < 0.02; Figure 7J) and the TA EMG burst was prolonged from 120.8 ± 19.61 ms to 179.3 ± 76.91 ms after 1 h (F_(4,36)_ = 3.136, *p* = 0.026; *Tukey’s multiple comparisons p* < 0.05; Figure 7L). 

Our analysis demonstrated significant changes in step cycle and EMG burst durations of the Sol and TA muscles in both groups of rats, grafted and non-grafted. Considering that in spinal non-grafted rats, the central presynaptic α_2_ adrenergic receptors are eliminated by total spinal cord transection [7,25], while our grafting procedure restores the NA innervation and their axons with α_2_ adrenergic receptors in the grafted rats, we suggest that some features of these step cycle indices might be controlled by presynaptic as well as postsynaptic α_2_ adrenergic receptors. 

#### 2.4.2. Relationship of the Burst Duration vs. Step Cycle Duration

It is known that during sustained locomotor performance the EMG activity of the flexor and the extensor muscles is characterized by an alternating pattern and that this activity pattern is responsible for the production of the alternating swing and stance phase of the consecutive step cycles. The duration of the EMG burst of the extensor muscle (Sol) activity is positively related to step cycle duration, while the duration of the EMG burst of the flexor TA muscle remains unchanged in relation to step cycle duration. Using the regression line analysis, which is a standard procedure in studies on locomotion to establish a normal locomotor pattern [26,27,28], we performed an additional analysis regarding the relationship between the EMG burst of the Sol and TA vs. step cycle durations. In the pre-drug situation locomotor performance of spinal grafted rats resembled the pattern of burst activity of the Sol and TA muscles obtained in intact rats (see our previous papers [11,24,29]). We observed that the duration of the EMG burst of the Sol muscle was positively related to the step cycle duration, and the regression line describing the relationship was characterized by a significant slope and a high correlation coefficient, while the duration of the EMG burst of the flexor TA muscle remained unchanged in relation to the step cycle duration, and the relationship was described by a flat regression line with a slope close to zero and low correlation coefficient (Figure 8A1,B1). Yohimbine treatment did not alter these relationships (Figure 8C1,E1), while clonidine altered the positive relationships of the Sol EMG burst vs. step cycle duration (Figure 8D1,F1). *RM ANOVA* confirmed a significant reduction of regression line slopes of the Sol EMG burst vs. step cycle durations after clonidine application (F_(5,55)_ = 3.885, *p* = 0.0044; *post-hoc Tukey’s multiple comparisons test p* < 0.05 for both time points) as well as an absence of a significant effect of yohimbine treatment (*p >* 0.05). The statistical analysis did not show any significant changes after yohimbine nor clonidine application in regression line slopes of the TA EMG burst vs. step cycle durations (F_(5,55)_ = 1.012, *p* = 0.4193).

In the case of control non-grafted rats, the relationships of the EMG burst duration of the Sol and TA muscles vs. step cycle duration were not affected by either yohimbine or clonidine. Please note that the Sol EMG burst activity was not positively correlated with the step cycle in pre- and after administration of either drug in non-grafted rats (Figure 8A2–F2). The statistical analysis did not show any significant changes after yohimbine nor clonidine applications in the regression line slopes of the Sol EMG burst vs. step cycle durations (Figure 8G2; F_(4,36)_ = 1.117, *p* = 0.3636) nor TA EMG burst vs. step cycle durations (Figure 8H2; F_(4,36)_ = 2.874, *p* = 0.0365; *post-hoc Tukey’s multiple comparisons test p* > 0.05 for all experimental conditions).

Our investigation shows that the slope of the regression line of the relationship between the Sol EMG burst duration vs. step cycle duration was affected by clonidine in grafted spinal rats. Such effect was not obtained in spinal non-grafted rats demonstrating that positive relationships of extensor burst duration vs. step cycle duration might be controlled by presynaptic α_2_ adrenergic located on neurons of the graft origin, which are eliminated by total spinal cord transection in spinal non-grafted rats [7,25].

#### 2.4.3. Interlimb Coordination 

Next, we tested the role of α_2_ adrenergic receptors on interlimb (left-right) coordination. For the polar plot analysis, we choose the same episodes of rhythmic EMG activity selected for step cycle, burst duration, and regression line analysis. In the pre-drug conditions, the locomotor performance recovery in spinal grafted rats was characterized by strong interlimb coordination (Figure 9A,D). Yohimbine application slightly improved the regularity of interlimb coordination (Figure 9B,C), while clonidine dramatically eliminated it (Figure 9E,F). Statistical analysis demonstrated a significant reduction of the strength of interlimb coordination (*r*-value) in grafted rats after clonidine application (F_(5,25)_ = 13.92, *p* < 0.0001; *post-hoc Tukey’s multiple comparisons test p* < 0.001 for both time points), while an effect of yohimbine treatment on the strength of interlimb coordination was not statistically significant (*p >* 0.05). At the same time, the strength of interlimb coordination in grafted rats after clonidine application was significantly lower in comparison to the strength of interlimb coordination observed in the same grafted rats during the locomotor performance before and after the yohimbine treatment (*post-hoc Tukey’s multiple comparisons test p* < 0.01 for pre-drug and *p <* 0.001 for the two-time points after). 

In spinal non-grafted rats, both drugs slightly worsened the already very limited locomotor performance. However, polar plot analysis did not show any significant deterioration or improvement of interlimb coordination by application of either drug in spinal control non-grafted rats (Figure 9G–L; *RM ANOVA* F_(4,16)_ = 1.854, *p* = 0.1679).

Our investigation demonstrates a significant role of noradrenergic innervation in control of hindlimb locomotor performance of spinal grafted rats expressed in control of interlimb coordination, step cycle, and burst durations due to α_2_ adrenergic receptors. In control spinal rats, pharmacological manipulation using the same as in grafted spinal rat doses of clonidine and yohimbine did not induce any positive effects. Overall, these results confirm the importance of noradrenergic innervation (in addition to serotonergic) for recovery of hindlimb locomotion in spinal grafted rats.

## 3. Discussion

Our data demonstrate that activation of α_2_ adrenergic receptors by clonidine (α_2_ agonist) abolished locomotor performance that was enhanced by intraspinal grafting in adult spinal rats. Inhibition of this receptor population by yohimbine (α_2_ antagonist) improved hindlimb locomotion substantially. Considering that yohimbine treatment enhances the release of NA while clonidine reduces it due to an action of these two drugs on α_2_ adrenergic auto-receptors [30,31], our results demonstrate that noradrenergic innervation plays a role in locomotor recovery in paraplegic grafted rats, in addition to enhancement provided by serotonergic innervation described in the past [9,10,11,12,13]. A role of NA neurons in such grafts was suggested after grafting pure 5-HT neuron suspension when NA neurons were eliminated using 6-OHDA, resulting in less effective locomotor recovery than that produced by a grafted suspension without the toxin [15]. Our investigations, for the first time, demonstrate the presence of several NA neurons in the graft harvested from the B1-B3 area, in addition to 5-HT neurons. We also found that the 5-HT and NA neurons were surrounded by each other’s axons demonstrating the presence of reciprocal connections between them in the graft. Moreover, we found that the NA innervation of the spinal cord was less pronounced than that of 5-HT and limited to specific areas of the spinal cord. 

In addition, we found that the 5-HT and NA neurons were postmitotic and differentiated at the time of embryonic tissue dissection for grafting and that the host environment did not stimulate their proliferation and differentiation. With high probability, the NA grafted neurons belong to the group of AC1-AC3 neurons (noradrenergic/catecholaminergic) of the caudal brainstem (see [8,16]), which are located in close proximity to the B1-B3 groups of the 5-HT neurons and were harvested for grafting along with the adjacent 5-HT neurons.

In spinal control non-grafted rats, we found only very few endogenous NA neurons in the spinal cord below total transection (not shown). This is consistent with previous work by others [32,33,34]. The very small number of these endogenous cells explains the relatively small effect of adrenergic α_2_ receptor manipulations in the locomotor performance of rats without a graft. Thus, we can conclude that the yohimbine related enhancement and clonidine related deterioration of hindlimb locomotor performance obtained in our grafted spinal rats was due to the graft origin NA neurons and their interconnection with graft origin 5-HT neurons**.**

### 3.1. Adrenergic α_2_ Receptors in Motor Control in Normal and in Grafted Rats 

The 5-HT/NA mechanisms involved in hindlimb movement control in grafted paraplegic rats might be explained by presynaptic or postsynaptic functional abilities of α_2_ adrenergic receptors. Guyenet with co-workers demonstrated that adrenergic neurons (AC1/AC3) including those with demonstrable spinal projections had detectable amounts of α_2_ adrenoceptors in the adult brainstem of normal rat [21]. Administration of clonidine inhibits the spontaneous firing of brainstem NA neurons as well as the firing of the great majority of 5-HT neurons in the midbrain [35]. The inhibition of NA cell firing in locus coeruleus and reduced NA release can be explained by the auto-receptor function of α_2_ adrenoceptors that are located on NA cell bodies and their axons [30]. Moreover, yohimbine increases the release of NA from neurons of locus coeruleus [31]. We propose that a similar mechanism to that obtained in intact rats may account for clonidine inhibitory and yohimbine facilitatory actions on spinal locomotion in our experiments.

In addition to this presynaptic action, a postsynaptic action of α_2_ adrenergic receptors can be considered on 5-HT neurons. There are solid data showing the presence of postsynaptic α_2_ adrenergic receptors within the rat brainstem [21,36]. Guyenet with co-workers, using a triple-label approach, found that all medullary serotonergic cells in the raphe pallidus, raphe obscurus as well as para-pyramidal (PPR) area including those with identified spinal projections contain punctate α_2_ adrenoceptors [21]. Thus, 5-HT release and 5-HT neuronal firing could be reduced via α_2_ adrenoceptors (coupled to the Gi/o pathway) located at the soma or presynaptic terminal of 5-HT neurons themselves, similarly to such modulation of 5-HT release in rat dorsal and median raphe nuclei described in the brainstem slices [37,38]. However, this issue needs further investigation.

In intact conditions, the brainstem noradrenergic and serotonergic structures control the activity of the Central Pattern Generator (CPG) spinal network as well as the activity of MNs. One among many neuronal populations constituting the spinal CPG network are commissural interneurons that coordinate left-right hindlimb muscle activity. Iontophoretic application of clonidine resulted in a depression of the commissural interneuron activity in the cat [39]. One can consider that in the rat an action of spinal interneurons (with commissural interneurons among them) is controlled similarly. Our grafting restores monoaminergic descending modulation of the CPG circuitry in the host spinal cord which may directly control interlimb coordination. This is consistent with our finding that interlimb coordination is defective after clonidine application. Such effect may rely on commissural interneurons as well as on other interneurons constituting the CPG network and in this way the grafted neurons may indirectly modulate the action of MNs. Our results do not deliver an answer to this question and it needs to be further investigated.

Noradrenaline also directly controls spinal motoneuron excitability as well as sensory afferent transmission to motoneurons in the spinal cord in normal rats and cats [40,41,42]. A direct role of α_2_ adrenergic receptors on MN activity is supported by morphological data showing that these receptors are detected in the spinal cord in the dorsal horn and in the part of the ventral horn containing motoneurons in rats as in cats [43,44]. The α_2_ adrenergic receptors localized on MN cell bodies might be responsible for the deterioration of hindlimb locomotor abilities after intrathecal application of yohimbine in intact cats [25] and intact rats [45]. However, α_2_ adrenergic receptor agonists such as clonidine also control MN action indirectly by inhibiting afferent transmission to motoneurons and directly by reducing motoneuron excitability due to reduced NA release by auto-receptor activation [40,46]. A review of the effects of NA and 5-HT on the control of afferent input is beyond the scope of this paper, but clearly clonidine and yohimbine influence afferents providing feedback during locomotion [47,48,49,50]. NA/5-HT neurotransmitters control CPG/MNs by interfering with afferent inputs from the hindlimb proprioceptors [39,40,46,51,52]. However, such direct NA modulation of MN activity is probably not present in our grafted rats due to rather limited restoration of NA innervation in the dorsal and ventral horn areas.

All these data discussed above support our proposal (see Conclusion) regarding the mechanism of an indirect NA action through 5-HT neurons in the graft and direct action of the NA descending system on the spinal cord CPG interneurons as well as indirect action on spinal MNs, which was replaced by intraspinal grafting in paraplegic rats.

### 3.2. Adrenergic α_2_ Receptors in Motor Control of Chronic Spinal Rats

Here we also investigated whether interfering with α_2_ adrenergic receptors using yohimbine and clonidine application could influence the locomotor abilities of the spinal rats without a graft. Our results are the first that demonstrate no positive effect on the locomotor performance of spinal chronic rats after administration of clonidine or yohimbine. After clonidine IP applications we observed some deterioration of hindlimb movements in spinal rats, which already presented very limited hindlimb movements in the control pre-drug experiments. This is in contradiction to results described in acute spinal cats, where clonidine could trigger robust and sustained hindlimb locomotion in the first week after the spinalization at a time when the cats were paraplegic, while yohimbine had no effect even at the largest doses [25]. In chronic spinal cats the effect of clonidine was less pronounced and although α_2_ adrenergic agonist markedly increased the cycle duration and flexor muscle burst duration, the weight support and extensor activity decreased as well as the foot drag was augmented [53]. In contrast to that, in cats with partial spinal cord lesions (ventral or ventrolateral part of the low thoracic spinal cord), clonidine application substantially deteriorated locomotor performance, while yohimbine application reversed this effect [7]. Rossignol with co-workers suggested that the clonidine related deterioration of locomotion of partial spinal cats, and clonidine beneficial effects in the early spinal cats, as well as the dose-dependent effects of clonidine application in late spinal cats with total transection could be explained by a difference in the α_2_ receptor population remaining in each experimental condition. A total spinal transection results in a removal of the central presynaptic α_2_ receptors [25] and the positive locomotor effect of clonidine application might rely on activation of the remaining postsynaptic receptors [39,54,55]. Some explanation for such differences of clonidine effects in intact vs. spinal rats was brought by Kehne and co-workers [56], who hypothesized that after spinal cord transection that eliminates central presynaptic α_2_ adrenergic receptors, a swing from α_2_ adrenergic mediated inhibition in normal condition to α1 adrenergic mediated excitation after spinalization might be obtained and concluded that the spinal transection unmasks clonidine’s α1 adrenergic stimulatory effect.

In intact cats intrathecal yohimbine application induced major walking difficulties characterized by asymmetric stepping, stumbling with poor lateral stability [25]. In intact adult rats intrathecal yohimbine application totally abolished locomotor performance [45], while in spinal rats investigated in our paper hardly any effect was observed. Our results are in opposition to data obtained from mice with total spinal cord transection, in which yohimbine markedly facilitated locomotion [57,58]. Clonidine could not acutely induce hindlimb movements in untrained and otherwise non-stimulated (e.g., neither tail nor perineal pinching) chronic spinal mice, and moreover occasionally occurring hindlimb movements normally obtained in a few days after spinal cord injury were acutely suppressed by clonidine application [58]. These authors propose that the suppressive effect of clonidine is mediated rather by the I1-imidazoline than by α_2_ adrenergic receptors what can be supported by comparable affinity of clonidine to both α_2_ and I1 receptors (*k*_i_ = 31.62–47.00 nM vs. *k*_i_ = 31.6228 nM; https://pdsp.unc.edu/databases/kidb.php). Also, in the yohimbine effect the other set of receptors (i.e., 5-HT_1A_, D_2_) might be involved when considering rather significant yohimbine affinity to them (*k*_i_ = 125.89 nM vs. *k*_i_ = 339 nM respectively). Obviously, the role of these types of receptors in conducting the clonidine or yohimbine modulation of locomotor performance might be considered but it is beyond the scope of our investigation and needs further clarification.

These results show clearly that the effects of noradrenergic pharmacotherapy (such as clonidine or yohimbine application) may depend on whether or not the spinal lesion is complete, whether it is acute or chronic state, as well as sex differences and the interspecies differences, have to be considered. Further work is needed to clarify this issue. 

## 4. Material and Methods

WAG (Wistar Albino Glaxo inbred strain) female rats 3-months-old with the bodyweight 165 ± 8 g at the beginning of the experiments were used in our investigations (*n* = 11). Experimental procedures and all surgical actions were carried out with care to minimize pain and suffering of animals, and were approved by the First Ethics Committee for Animal Experimentation in Poland (decision no. 303/2017 and no. 753/2018), according to the principles of experimental conditions and laboratory animal care of the European Union (EU Directive, 2010/63/EU) and the Polish Law on Animal Protection. 

### 4.1. Complete Spinal Cord Transection

Complete spinal cord transection (SCI) was performed in aseptic conditions under deep anesthesia (Isoflurane, 2% and Butomidor, 0.05 mg/Kg b.w.). First, a mid-dorsal skin incision was performed over the Th8-Th11 vertebrae at the rat back. Next, the back muscles were separated from the vertebral spines. After laminectomy between the Th9/Th10 segments, the spinal cord was completely transected. A piece of spinal cord tissue was gently aspirated after two cuts performed with scissors at a 1–2 mm distance to prevent any axonal regrowth through the cavity of the lesion, as previously described [10,11,12]. After stopping hemorrhage in the lesion cavity, the para-vertebral muscles and fascia were closed in layers using sterile sutures. Then, the skin was closed using stainless-steel surgical clips. After surgery, the animals received a single dose of non-steroidal anti-inflammatory and analgesic treatment (Tolfedine 4 mg/Kg subcutaneously; s.c.) and antibiotics for the following 4 days (Gentamicin 10 mg/Kg s.c.). For about 7 days after spinal cord injury, until the voiding reflex was re-established, the bladder was emptied manually twice a day. Female rats were chosen for these experiments due to relatively fewer problems with the bladder emptying in acute time after spinal cord injury.

### 4.2. Intraspinal Grafting of 14-Day Old Rat Embryo Brainstem

One week after spinal cord transection, 6 out of 11 spinal rats were selected randomly for intraspinal grafting of embryonic brainstem tissue as previously described [10]. To label any proliferating fetal cells, time-pregnant female WAG rats were treated with 5-Ethynyl-2′-deoxyuridine (EdU; ThermoFisher Scientific, Waltham, MA, USA). The injections were performed three times at a dose of 50 mg/Kg, on embryonic day 13 (E13). The embryos were removed by cesarean section at the next day (E14) and transferred to Hanks’ buffered solution. The embryonic raphe B1-B3 area, approximately 0.25 × 0.5 × 1 mm, was dissected under a surgical microscope. For grafting procedure, a sharpened glass micropipette connected to a Hamilton syringe was used. A dissected solid piece of embryonic tissue was injected into a host rat spinal cord 1 mm below the pial surface through a small laminectomy performed at the T10/11 vertebrae level of the host spinal cord, which corresponds to the Th12 thoracic spinal cord level (Figure 10). Care was taken for a slow withdrawal of the glass pipet to avoid leakage and secure the graft within the spinal cord after transplantation.

### 4.3. Implantation of EMG Recording Electrodes

Two to three months after intraspinal grafting the animals were anesthetized with isoflurane anesthesia (5% to induce and then maintained with 2% in oxygen 0.2–0.3 L/min) and bipolar EMG recording electrodes were implanted to the selected hindlimb muscles, as previously described [10,11,12]. The soleus (Sol) and tibialis anterior (TA) muscles, representing flexor and extensor groups of hindlimb muscles were chosen to assure the possibility of alternating rhythmic EMG activity recordings during hindlimb locomotor movement investigations. The homemade EMG electrodes were produced with Teflon-coated stainless-steel wire (0.24 mm in diameter; AS633, CoonerWire Co., Chatsworth, CA, USA). The hook electrodes, with the insulation at the tip removed (0.5–1 mm), were implanted with a distance of about 1–2 mm in the chosen muscle and secured by a suture. The ground electrode was secured on the back muscles under the skin in some distance from the hindlimb muscles. The other ends of the EMG electrodes were fixed to the connector, covered with dental cement (Spofa Dental, Prague, Czechia) and silicone (3140 RTV, Dow Corning, Midland, MI, USA), which was secured by suture under the skin to the back muscles of the animal. This method does not produce any discomfort of investigated rats during spontaneous movement, so allows chronic EMG activity recordings from selected muscles for several weeks in our paraplegic rats during testing locomotor hindlimb muscle activity.

### 4.4. Video and Electromyographic Recordings

One week after EMG electrodes implantation, the muscle activity pattern related to the hindlimb locomotor movement was investigated in rats placed with their forequarters and forelimbs on a platform above a treadmill (Panlab/Harvard Apparatus, USA) with their hindlimbs touching the moving belt (10 cm/s). Locomotor-like hindlimb movement was induced by the tail pinching as described previously [9,10,11,12,15]. The EMG activity recording was synchronized with video recordings to enable the behavioral offline analysis. The filtered (0.1 to 1 KHz bandpass) and digitized EMG activity was stored on a computer (3 KHz sampling frequency) and analyzed offline using the Winnipeg Spinal Cord Research Centre data capture and analysis software (http://www.scrc.umanitoba.ca/doc/). First, the raw EMG was rectified and integrated with 5 ms interval. Next, marking all the burst onsets and offsets in all the EMG activity recorded was performed. Based on marked burst onsets and offsets we established the cycle duration (time between two consecutive EMG bursts), burst duration (time between the onset and offset of the EMG burst) in different rats for different experimental conditions. The interlimb coordination between the TA muscles of the left and right hindlimbs was determined using polar plot analysis [59,60,61,62]. In the polar plots, the position of the vector reflects the phase shift between the analyzed EMG burst onsets; 0 or 360° corresponds to synchrony, while 180° is equivalent to alternation. The strength of coordination between analyzed muscle burst onsets corresponds to the length of the ***r***-vector, ranging from 0 to 1, which is related to the concentration of phase values (single dots at the polar circle) around the mean. When the value of *r*-vector is close to 1 and the individual dots illustrating the phase shift between analyzed EMG bursts are concentrated around the mean, the onsets of analyzed EMG bursts are strongly coupled. Conversely, when bursting in analyzed muscle burst activity is independent and there is no coupling, the phase distribution shows some degree of dispersion, with a wide distribution of dots on the polar circle. In our analysis, to determine whether the interlimb coordination ***r***-values were concentrated (burst coupling) or dispersed (no burst coordination), Rayleigh’s circular statistical test was applied. The interlimb coordination was considered to be phase-related when the ***r***-value was greater than critical Rayleigh’s value (cR) for a given *P* value [61]. For a more detailed description see our recent paper [63].

Video recordings collected during locomotor testing were used to establish and compare the number of proper plantar steps performed by investigated animals during 40 s recordings before and at time points after drug application. A proper plantar step was defined as the left hindlimb step with obvious paw contact with the ground and substantial body weight support, with no toe dragging, and followed by a proper step cycle of the contralateral hindlimb. Two investigators (one blind) independently analyzed data offline by inspecting the video recording carried out during the experimental procedure.

### 4.5. Evaluation of Hindlimb Locomotion

The plantar stepping was evaluated based upon EMG analysis, as described previously [11,29]. The rats placed with their forequarters and the forelimbs on a platform above the moving belt and tail pinch was used to elicit hindlimb movements. Stimulation of tail or perineal area afferents is a method commonly used for eliciting locomotion in spinal rats and cats [29,64,65,66], and in many prior attempts to reveal locomotor recovery in paraplegic rats with the intraspinal embryonic tissue grafting [9,10,11,12,15]. To maximize the quality of plantar stepping of grafted spinal rats the tail stimulus was adjusted by the experimenter in all experimental conditions. The characteristic features of proper plantar stepping were considered as follows: sustained EMG burst of soleus activity during the stance phase, which duration was related to step cycle duration, as well as a brief TA EMG burst activity of consistent duration, and in addition consistent interlimb movement associated with alternating EMG burst activity of left and right as well as of flexor and extensor EMG burst of homonymous muscles of both hindlimbs.

### 4.6. Drug Tests

To investigate whether the noradrenergic (NA) innervation plays functional role in the control of locomotor hindlimb movements recovered by intraspinal grafting of embryonic brainstem tissue in spinal rats, we analyzed locomotor hindlimb abilities in paraplegic rats after application 0.2–0.3 mg/Kg (intraperitoneally; IP) of clonidine (α_2_ adrenergic receptor agonist) or 0.5 mg/Kg (IP) of yohimbine (antagonist of the α_2_ adrenergic receptor). Following a pre-drug trial, the effects of an α_2_ adrenergic receptor agonist or antagonist on locomotor performance were tested 5–30 min after drug injection (i.e., when the clear effect of the drug administration was usually observed) and then 1 h later. To avoid any training or drug accumulation effects the locomotor testing was carried out not more often than twice a week separated by 3 days-intervals.

### 4.7. Immunohistochemistry

After completing the functional investigations and EMG recordings, the grafted and non-grafted spinal rats were subjected to perfusion under deep anesthesia induced by intraperitoneal injection of sodium pentobarbital (150 mg/Kg). The process involved cold 0.1 M PBS (phosphate buffered saline, pH = 7.2) and cold 4% paraformaldehyde in PBS, for 3 and 15 min, respectively. Cryoprotection of the spinal cords was ensured with 30% sucrose in PBS. Afterward, spinal cords were embedded in optimal cutting temperature compound (OCT embedding matrix, CellPath, UK) and frozen on dry ice. For immunohistochemistry, it was cut into 12 µm slices and immobilized on glass covered with poly-L-lysine. Frozen sections were rinsed in PBS and blocked in 10% normal donkey serum in PBS with 0.5% Triton X-100 at RT for 2 h. Overnight incubation with primary antibodies at 4° C was followed by treatment with fluorophore-conjugated secondary antibodies for 2 h at RT. After washes in PBS, the sections were covered with fluorescence mounting medium (Dako, Agilent, Santa Clara, CA, USA) and examined with the aid of a Zeiss Fluorescence Microscope Axio Imager M2. The immunohistochemistry involved rabbit anti-tyrosine hydroxylase (1:1000; Abcam, Cambridge, UK), sheep anti-tryptophan hydroxylase (1:200; Merck, Darmstadt, Germany) and goat anti-5-HT (1:1000; ImmunoStar, Hudson, WI, USA) primary antibodies, as well as the Alexa Fluor secondary antibodies: 488- and 555-conjugated donkey anti-rabbit/goat/sheep IgG (1:1000; ThermoFisher Scientific, Pittsburg, PA, USA). Nuclei were dyed with 1 μg/mL DAPI in H_2_O (Sigma-Aldrich). Visualization of EdU positive cells was obtained with Click-iT EdU Cell Proliferation Kit for Imaging (ThermoFisher Scientific, Pittsburg, PA, USA). For cell counting, serial micrographs were acquired from each fifth section per graft area, each of 70 µm apart. 

### 4.8. Statistical Analysis

All values in our paper are reported as mean **±** SD (Standard Deviation). For statistical comparison of various parameters in the different experimental conditions, *two-way Repeated Measures Analysis of Variance* (*RM ANOVA*) with *Tukey’s test for multiple comparisons* was used (GraphPad Software version 8.1.1 for Windows, San Diego, California USA, www.graphpad.com). The minimum significance was set at *p* = 0.05. For individual muscles, the relationship between burst duration vs. cycle duration was determined using regression line analysis [26,27]. 

## 5. Conclusions

As we described above our morphological data show the 5-HT neurons surrounded by NA axons as well as the NA neurons surrounded by 5-HT axons in the graft region suggesting functional synaptic connections between them or at least volume transmission might be considered. Since our results show that clonidine abolished locomotor performance in spinal grafted rats, while yohimbine improved it, we can conclude that α_2_ adrenergic receptors are involved in the action of 5-HT/NA systems in the locomotor recovery of spinal grafted rats. Our results and literature data allow us to propose direct and indirect mechanisms of the 5-HT and NA interaction that can be responsible for the locomotor recovery after our intraspinal strategy (see Figure 11).

First, there is a direct action of the 5-HT descending system on the spinal cord CPG/MNs that was replaced by intraspinal grafting. Our previous investigations using antagonists of defined 5-HT receptors demonstrated the important role of 5-HT_2_ and 5-HT_7_ receptors in the recovery of hindlimb locomotion (the green descending pathway in Figure 11; for details see [11,12]). Secondly, there is a possible direct action of the NA descending system that was replaced by intraspinal grafting on the spinal cord CPG/MNs. Our data showed the involvement of α_2_ adrenergic receptors on the hindlimb locomotor performance in the spinal rats restored by the graft. There are solid data in the literature describing the descending NA pathways from locus coeruleus targeting in the dorsal horns and the ventral horn around motoneurons as well as around the central canal and in the intermediolateral zone indicated by the red (field and dashed) arrows in Figure 11 [9,67]. However, in our investigations, we observed re-established innervation mainly around the central canal and in the intermediolateral zone but hardly any fibers were detected in the ventral horn around MNs (see in Figure 11 the descending pathway marked with red field arrow in contrast to dashed red arrows indicating the other descending NA pathways in normal rats which were not re-established by our grafting strategy). The NA grafted neurons belong to the group of AC1-AC3 neurons (noradrenergic/catecholaminergic) of the caudal brainstem (see [20]), which are located in close proximity to the B1-B3 groups of the 5-HT neurons and were harvested for grafting along with the adjacent 5-HT neurons. These NA neurons are not part of locus coeruleus and this might be a reason for the absence of terminals in the area normally innervated by the locus coeruleus. 

The next possible NA/5-HT actions involved in motor control are indirect pathways involving NA control of 5-HT release [68] and pathways involving 5-HT control of NA release [69,70]. The important elements of indirect pathways controlling hindlimb movement might be related to the reciprocal connections between 5-HT and NA neurons within the graft that in addition to 5-HT receptors (not shown in the figure) are mediated by α_2_ adrenergic receptors (see Figure 10). The 5-HT fibers present in the ventral horns around the MNs might be a part of indirect pathway of noradrenergic locomotor control re-established by our grafting strategy. As the 5-HT fibers are robust also in the dorsal horns the indirect control of sensory feedback from the hindlimb by NA should be also considered, but further work is required to define the afferents involved in this control.

Our grafting procedure supplies the spinal cord with the partly re-established brainstem 5-HT and NA systems, both of which contribute to locomotor recovery in paraplegic grafted rats. Whether the mechanisms that are responsible for locomotor recovery after intraspinal grafting remain those from the intact motor control system needs further investigations.

## Figures and Tables

**Figure 1 ijms-21-05520-f001:**
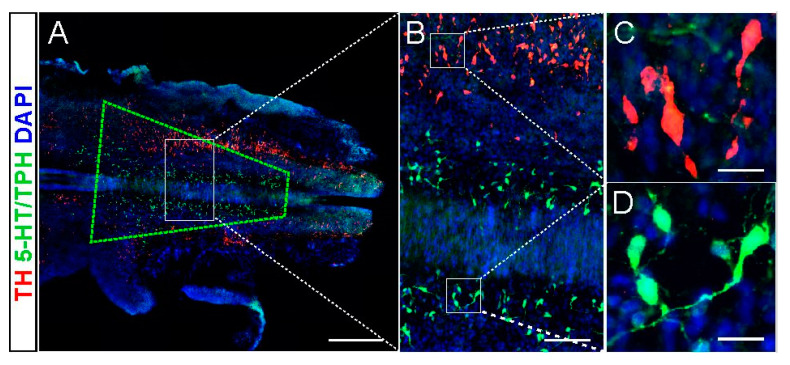
A representative section of the embryonic brainstem with the region of the tissue taken for grafting marked with green quadrangle (**A**). Selected areas are presented with higher magnifications (**B**–**D**). An example of 5-HT neurons (5-HT/TPH-positive; green) in the B1/B2/B3 groups (**D**) and in close lateral proximity NA neurons (TH positive; red) in the AC1/AC2/AC3 groups (**C**). For nucleic acid staining, DAPI was used (blue). Scalebars: 500 µm (A); 100 µm (B); 20 µm (C,D). Abbreviations: 5-HT: serotonin; TPH: tryptophan hydroxylase; TH: tyrosine hydroxylase DAPI:4′,6-diamidino-2-phenylindole.

**Figure 2 ijms-21-05520-f002:**
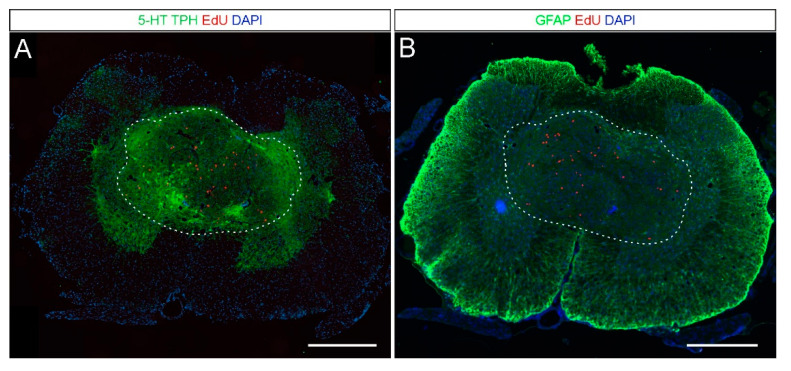
Representative immunofluorescent staining of two cross-sections from the same rat (about 180 µm apart) demonstrating the integration of the grafted tissue into the host spinal cord. (**A**) The location of the grafted tissue (marked by a dashed line) was estimated by the presence of 5-HT/TPH (green) positive neurons, their fibers, and EdU positive nuclei present exclusively in the cells of embryonic origin (red). Please note that several 5-HT/TPH-positive neurons are located close to the graft border and that there is the robust serotonergic fiber innervation of the spinal cord areas outside the graft region (i.e., the ventral horns). (**B**) There were no reactive astrocytes detected at the edge of the graft region (GFAP; green). Scalebars: 500 µm. Abbreviations: 5-HT: serotonin; TPH: tryptophan hydroxylase; EdU: 5-ethynyl-2-deoxyuridine; DAPI: 4′,6-diamidino-2-phenylindole; GFAP: glial fibrillary acidic protein.

**Figure 3 ijms-21-05520-f003:**
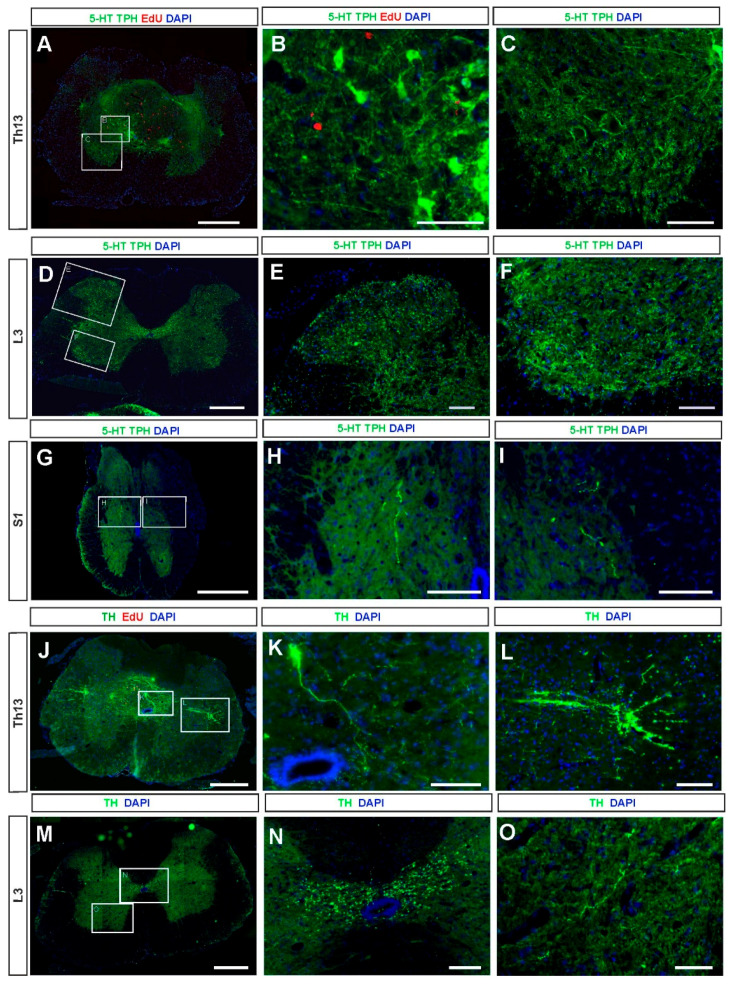
The 5-HT (**A**–**I**) and NA (**J**–**O**) neurons of graft origin and their axons innervating host spinal cord structures. Representative cross-section of the spinal cord at the epicenter of the graft (approximately the Th13 spinal segment) with serotonergic neurons (**A**; 5-HT/TPH-positive, green) and the graft area under higher magnification (**B**). Serotonergic fibers are present in the ventral horn (under higher magnification in **C**) and in the spinal cord cross-section located at the L3 lumbar spinal cord segment (**D**) as well as in the dorsal horn (**E**) and in the ventral horn (**F**) of spinal cord cross-section illustrated in D. The 5-HT fibers were found as far as 20 mm from the epicenter of the graft (presented in A) caudally in the S1 (proximately) host spinal cord (**G**–**I**). Noradrenergic (NA; TH positive, green) neurons and fibers are present in the epicenter of the graft (approximately the Th13 spinal segment) (**J**) and example of NA neuron (**K**) as well as NA fibers in the intermediolateral zone (**L**). NA fibers are present in the cross-section of the L3 lumbar spinal cord level (**M**) and under higher magnification in the area of the central canal (**N**) as well as in the ventral horn, however rather sparsely (**O**). For nucleic acid staining, DAPI was used (blue). Please note that there are no EdU (red) positive serotonergic or noradrenergic neurons in the graft region, confirming that these neurons were already postmitotic at the time of harvesting for grafting from the E14 embryo. Scalebars: 500 µm (A,D,G,J,M); 100 µm (B,C,E,F,H,I,K,L,N,O). Abbreviations: 5-HT: serotonin; TPH: tryptophan hydroxylase; TH: tyrosine hydroxylase; EdU: 5-ethynyl-2-deoxyuridine; DAPI: 4′,6-diamidino-2-phenylindole; Th13, L3, S1: the thoracic 13, lumbar 3, and sacral 1 spinal cord segments.

**Figure 4 ijms-21-05520-f004:**
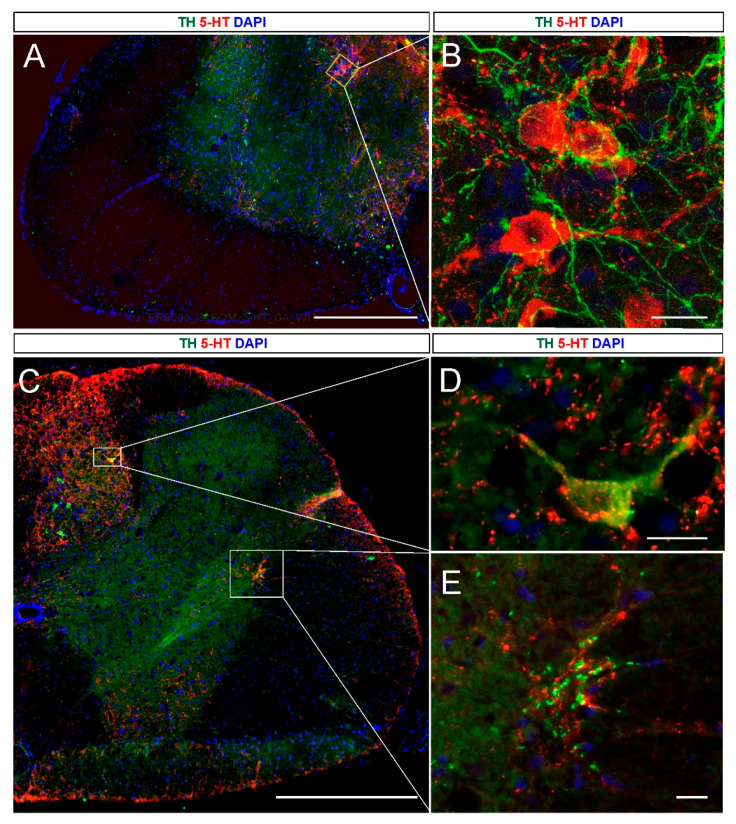
The 5-HT and NA neurons surrounded by each other’s axons identified in the graft vicinity in the host spinal cord of grafted rats. (**A**) an example of the graft area with 5-HT neurons (red) and NA (green) fibers on the cell membrane; (**B**) an example of selected neurons under higher magnification in the confocal image; (**C**) the graft area with 5-HT (red) and NA neurons (green); (**D**) an example of a selected neuron with 5-HT fibers on the cell membrane under higher magnification in the fluorescent image; (**E**) the intermediolateral zone rich 5-HT/NA innervation of graft origin under higher magnification in the fluorescent image. For nucleic acid staining, DAPI was used (blue). Scalebars: 500 µm (A,C); 20 µm (B,D,E). Abbreviations: 5-HT: serotonin; TPH: tryptophan hydroxylase; TH: tyrosine hydroxylase DAPI: 4′,6-diamidino-2-phenylindole.

**Figure 5 ijms-21-05520-f005:**
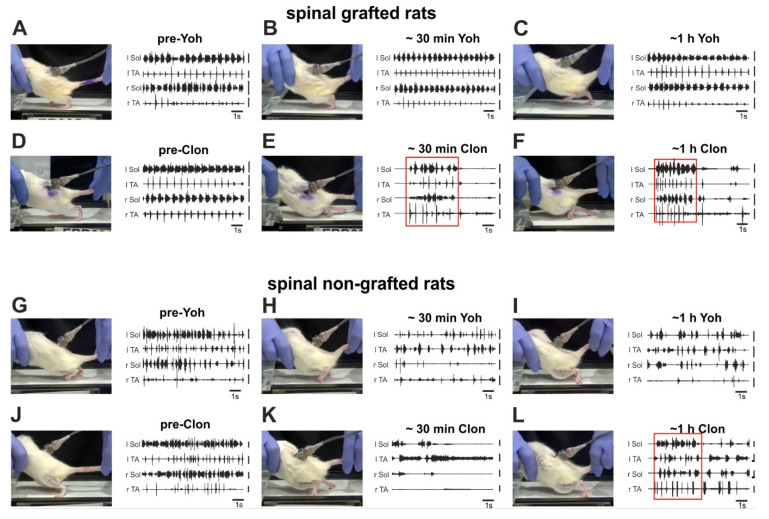
The top two panels present video frames of the spinal grafted rats with a typical posture and accompanied EMG activity recorded during the locomotor performance on the treadmill (**A**–**F**) before and after application of yohimbine (antagonist of α_2_ adrenergic receptors) that substantially improved locomotor hindlimb movements (**A**–**C**); before and after application of clonidine (agonist of α_2_ adrenergic receptors) resulting in deterioration of body weight support and both hindlimbs dragging over the moving treadmill belt (**D**–**F**). The bottom two panels (**G**–**L**) present video frames of the spinal non-grafted rats with a typical posture and accompanied EMG activity recorded during the locomotor performance on the treadmill before and after application of yohimbine (**G**–**I**) that was not accompanied by any visible changes in limb performance and before and after application of clonidine (**J**–**L**) resulting in substantial deterioration of already limited limb movements. Abbreviations: l: left; r: right; Sol: soleus muscle; TA: tibialis anterior muscle; ~30 min: about 30 min (short time) after drug application; ~1 h: about 1 h (long time) after drug application; Clon: clonidine; Yoh: yohimbine.

**Figure 6 ijms-21-05520-f006:**
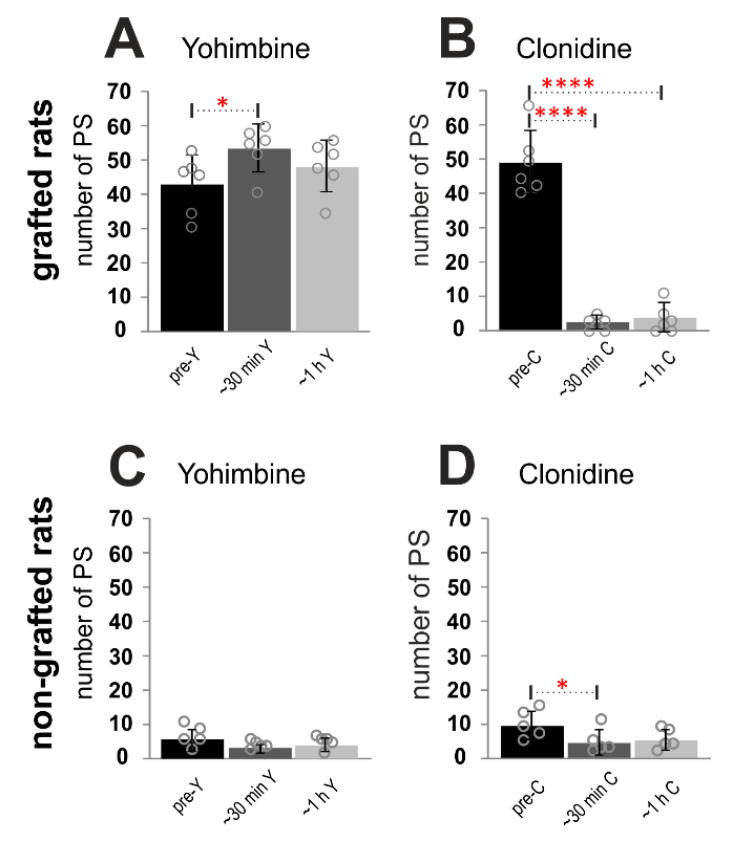
Number of proper plantar steps (PS) in 40 s test of the locomotor performance induced by tail pinching of spinal grafted (*n* = 6) rats (**A**,**B**) and spinal non-grafted (*n* = 5) rats (**C**,**D**) on the treadmill (10 cm/s) before and after application of yohimbine (**A**,**C**) or clonidine (**B**,**D**). The bars represent means ± SD established for the two groups animals (with and without a graft) in various experimental conditions, while gray circles present the data from individual rats. *RM ANOVA* with *Tukey’s multiple comparisons test*: * *p* < 0.05; **** *p* < 0.0001. ~30 min: < 30 min after drug application; ~1 h: one hour after drug application; C: clonidine; Y: yohimbine.

**Figure 7 ijms-21-05520-f007:**
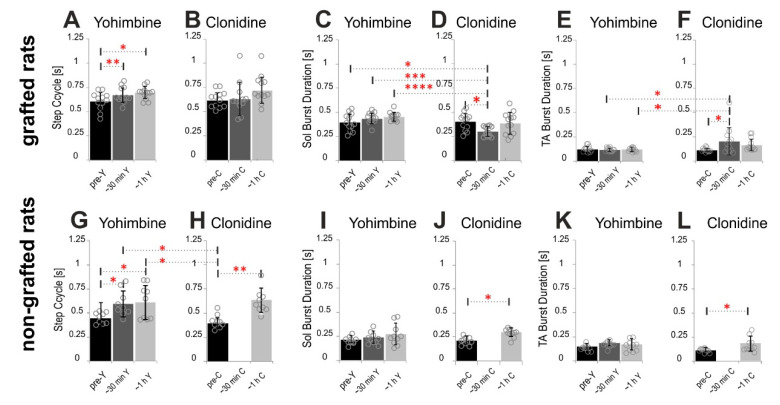
The cycle duration and EMG burst duration of the Sol and TA muscles before and after yohimbine or clonidine aplications in spinal grafted (*n* = 6) rats (**A**–**F**) and non-grafted (*n* = 5) spinal (**G**–**L**) rats as well as before and after clonidine in grafted (**D**–**F**) and non-grafted (**J**–**L**) spinal rats. The bars represent means ± SD established for the two groups of rats (with and without a graft) in various experimental conditions, while gray circles present the data from individual rats. *RM ANOVA with Tukey’s multiple comparisons test*: * *p* < 0.05; ** *p* < 0.01;*** *p* < 0.001; **** *p* < 0.0001. Abbreviations: Sol: soleus muscle; TA: tibialis anterior muscle; ~30 min: < 30 min after drug application; ~1 h: one hour after drug application; C: clonidine; Y: yohimbine.

**Figure 8 ijms-21-05520-f008:**
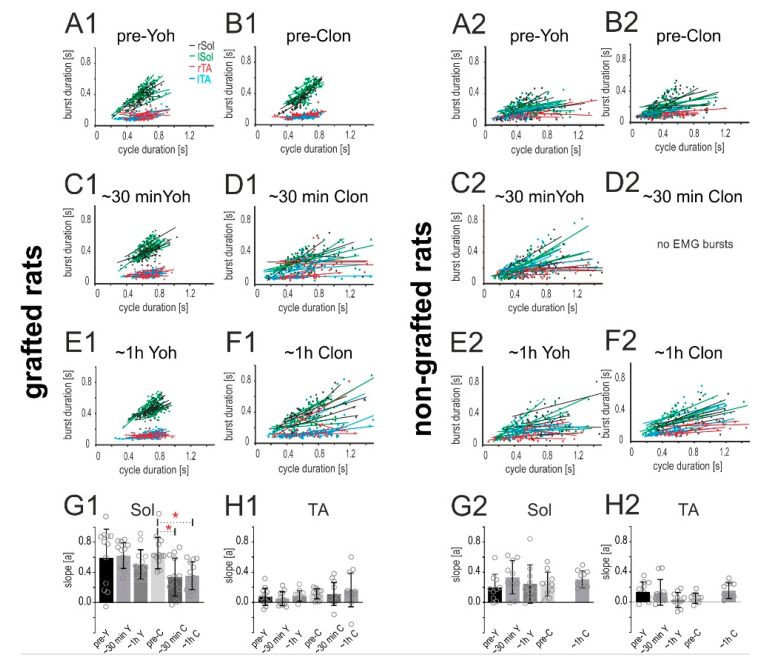
Relationships between burst duration vs. step cycle duration established for the Sol and TA EMG burst activity in both hindlimbs of spinal grafted rats (*n* = 6) after yohimbine (**A1**,**C1**,**E1**) and clonidine (**B1**,**D1**,**F1**) application as well as in spinal non-grafted rats (*n* = 5) after yohimbine (**A2**,**C2**,**E2**) and clonidine (**B2**,**D2**,**F2**) application. Each regression line established for the burst EMG activity of a given rat muscle is illustrated by a given color (green/black—left/right Sol; blue/red—left/right TA). The bars in **G1**, **H1**, **G2**, **H2** represent the slopes (means ± SD) of regression lines of the relationships between the burst duration vs. step cycle durations for the Sol and TA muscles respectively established for the two groups of rats (with and without a graft) in various experimental conditions, while gray circles present the data from individual rats. Note the locomotor deteriorations after clonidine application are associated with the significant decrease of regression line slopes in the case of Sol (G1) but not TA EMG (H1). *RM ANOVA with Tukey’s multiple comparisons* * *p* < 0.05. Abbreviations: Sol: soleus muscle; TA: Tibialis Anterior muscle; l: left; r: right; Clon, C: clonidine; Yoh, Y: yohimbine; ~30 min: < 30 min after drug application; ~1 h: one hour after drug application.

**Figure 9 ijms-21-05520-f009:**
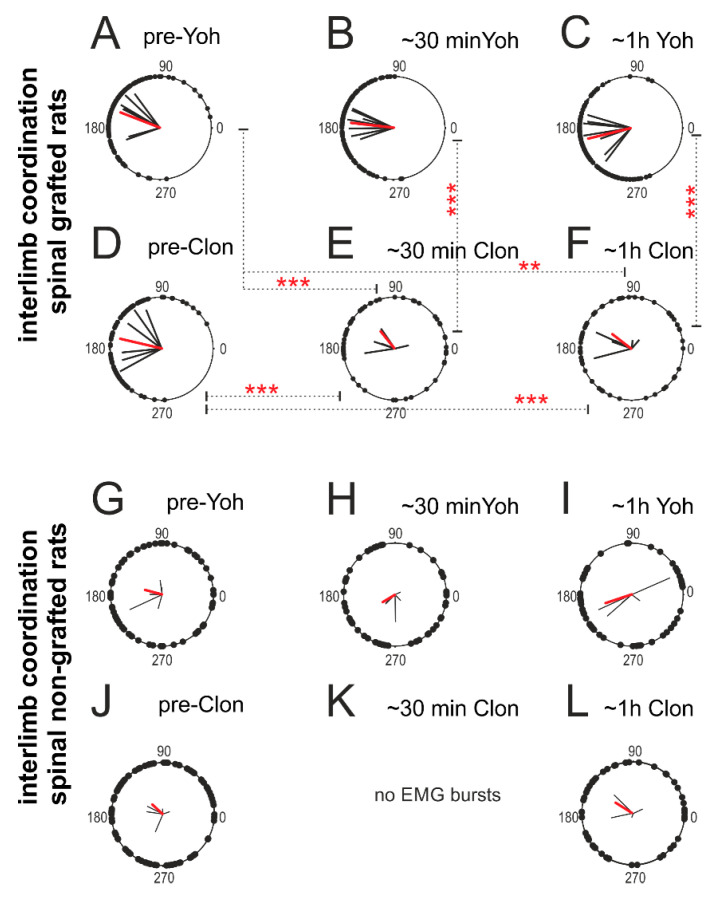
Interlimb coordination in locomotor performance (established between left vs. right TA muscles) in spinal grafted rats (*n* = 6) after yohimbine (**A**–**C**) and clonidine (**D**–**F**) application. Interlimb coordination in locomotor performance (established between left vs. right TA muscles) in spinal non-grafted rats (*n* = 5) after yohimbine (**G**–**I**) and clonidine (**J**–**L**) application. Black vectors are established for each pair of TA muscles in individual rats. Red vectors are the mean for the group of animals. Single dots in the polar plots illustrate the phase shift between analyzed onsets of EMG burst of left vs. right TA in individual step in all the analyzed animals. *RM ANOVA with Tukey’s multiple comparisons* ** *p* < 0.01; *** *p* < 0.001. Abbreviations: Clon—clonidine; Yoh—yohimbine; ~30 min—< 30 min after drug application; ~1 h—one hour after drug application.

**Figure 10 ijms-21-05520-f010:**
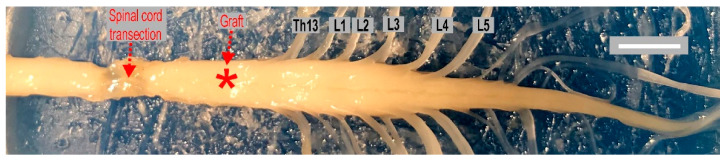
Spinal cord of adult WAG rat (Wistar Albino Glaxo inbred strain) with a clear scar in the site of total transection and an approximate location of a graft performed in our experiments. Scalebar: 5 mm. Abbreviations: Th13, L1-5 thoracic 13, lumbar 1–5 spinal cord segments.

**Figure 11 ijms-21-05520-f011:**
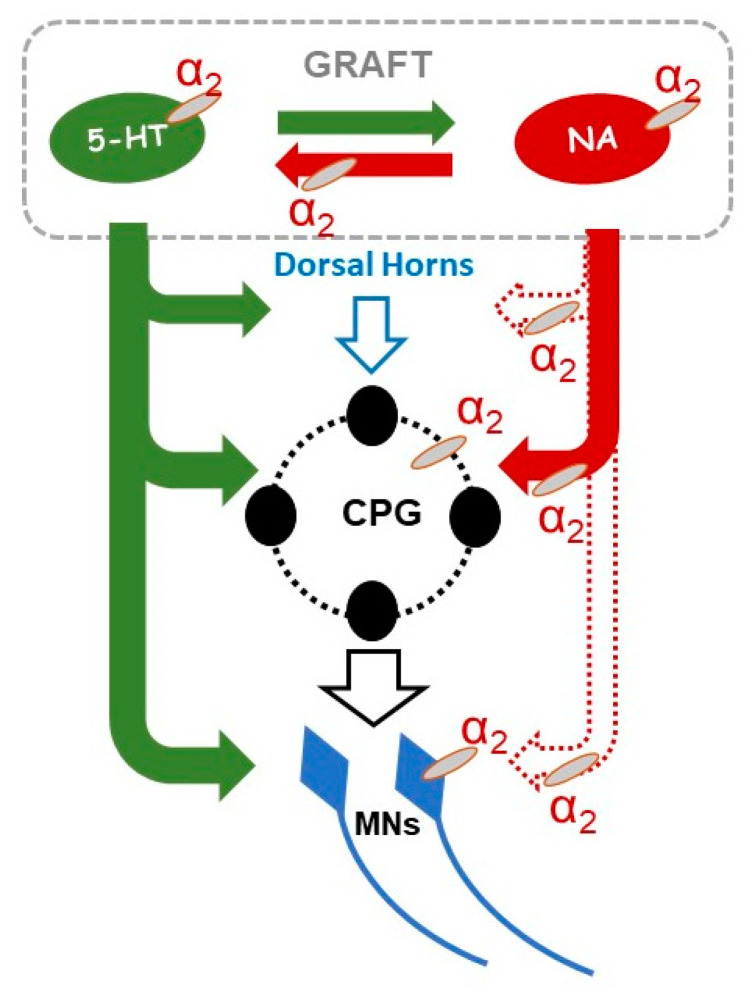
Schematic diagram with graphic indications of postulated serotonergic/noradrenergic (5-HT/NA) direct and indirect actions in the control of hindlimb locomotor performance in paraplegic rats with intraspinal grafting of the embryonic brainstem tissue. The 5-HT and NA axonal connections with dorsal horns and CPG structures and MNs in the ventral horns as well as a reciprocal interaction between 5-HT <=> NA indicate possible direct and indirect mechanisms involved in hindlimb locomotor performance: 1/ action via 5-HT descending system on the spinal cord CPG/MNs (green arrows); 2/ action via NA descending system on the spinal cord CPG/MNs (red arrows); 3/ action based on reciprocal connections of NA vs. 5-HT neurons within the graft (square NA <=> 5-HT) that controls the 5-HT and NA release (for more details see the Discussion). The red field arrow indicates NA connection established in this study while the red dashed lines indicate NA connections in control condition, not re-established by the intraspinal grafting in the present paper. Abbreviations: NA—tyrosine hydroxylase positive neurons in the graft; 5-HT—tryptophan hydroxylase/serotonin positive cells in the graft; MNs—motoneurons; CPG—central pattern generator; α_2_—α_2_ adrenergic receptors.

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
