# Peer review of "Noradrenergic Components of Locomotor Recovery Induced by Intraspinal Grafting of the Embryonic Brainstem in Adult Paraplegic Rats"

_ijms, 2020, doi:10.3390/ijms21155520_

Round 1

Reviewer 1 Report

The aim of this paper was to elucidate the role of NA innervation in locomotor recovery after Th9/Th10 complete spinal transection in female adult rats. About half of the injured animals received a graft of embryonic (E14) brainstem tissue containing B1-B3 5HT neurons and AC1-AC3 NA neurons. Both grafted and non-grafted cohorts were implanted with EMG electrodes in left and right hindlimbs for electrophysiological recording of treadmill locomotor activity. All animals were injected with clonidine and yohimbine to test the effects of alpha2 adrenergic receptors mechanisms on locomotor activity. The study is well thought out and the rational for the different sets of experiments are clear. Novel and interesting findings make an important contribution to the literature. Nevertheless, a few points should be addressed to clarify the results and their interpretation, and facilitate reading by a large audience.

Abstract

line17: …containing 5HT and NA neurons…

lines29-30: I would suggest mentioning the effects of these drugs on locomotor abilities of non-grafted injured rats as well.

Introduction

The authors should discuss the affinity of clonidine and yohimbine for other receptors than presumed presynaptic alpha2 adrenergic (e.g., postsynaptic adrenergic, 5HT and imidazoline). This should probably be done in Intro to help better understand statements such as those lines 270-273 and lines 347-350.

Results

section 2.2. Some quantification of the anatomical data described would greatly strengthen this section.

line151: “Contrary to our expectation, …”. Needs explanation.

Figures

Individual data points (from the 6 grafted and 5 not-grafted rats) should be added to all bar graphs.

Figures 4 and 5: For consistency with other figs. and to help better understand F stats reports, pre-drug conditions should be labeled “pre-Yoh or pre-Clon”.

I suggest merging Figs.7 and 8 into one single figure and Figs.9 and 10 into another.

Discussion

line 424: “…fully differentiated…” requires clarification.

lines 449-456: The postsynaptic scenario is difficult to follow.

line 461-462: after spinal cord injury? Also in ref 39 refers to clonidine induced depression of descending-evoked responses in commissural interneurons in intact cord.

line 512-515: could sex differences be included as well?

Methods

line 644: RM not defined

Conclusions

lines 650-651: suggest synaptic connection. These may or may not be functional.

lines 676-678: reciprocal connections are not supported by results shown.

Author Response

The aim of this paper was to elucidate the role of NA innervation in locomotor recovery after Th9/Th10 complete spinal transection in female adult rats. About half of the injured animals received a graft of embryonic (E14) brainstem tissue containing B1-B3 5HT neurons and AC1-AC3 NA neurons. Both grafted and non-grafted cohorts were implanted with EMG electrodes in left and right hindlimbs for electrophysiological recording of treadmill locomotor activity. All animals were injected with clonidine and yohimbine to test the effects of alpha2 adrenergic receptors mechanisms on locomotor activity. The study is well thought out and the rational for the different sets of experiments are clear. Novel and interesting findings make an important contribution to the literature. Nevertheless, a few points should be addressed to clarify the results and their interpretation, and facilitate reading by a large audience.

We are very pleased with the effort taken by the Reviewer. We have corrected our manuscript according to the Reviewer’s suggestions. We hope our corrections made the text of our manuscript improved and clear now.

Abstract

line17: …containing 5HT and NA neurons…

We have modified the first sentence (line 17) to make clear that the results were obtained previously and already published by us and others in the past.

lines29-30: I would suggest mentioning the effects of these drugs on locomotor abilities of non-grafted injured rats as well.

We have modified the abstract to make a clear statement (line 28/29) that:

Locomotor abilities of the spinal grafted rats, but not in control spinal rats were facilitated by yohimbine and suppressed by clonidine.

Introduction

The authors should discuss the affinity of clonidine and yohimbine for other receptors than presumed presynaptic alpha2 adrenergic (e.g., postsynaptic adrenergic, 5HT and imidazoline). This should probably be done in Intro to help better understand statements such as those lines 270-273 and lines 347-350.

We have modified the Discussion section according to the reviewer suggestion and now it is as follows (lines 508- 520):

“Our results are in opposition to data obtained from mice with total spinal cord transection, in which yohimbine markedly facilitated locomotion [57,58]. Clonidine could not acutely induce hindlimb movements in untrained and otherwise non-stimulated (e.g., neither tail nor perineal pinching) chronic spinal mice, and occasionally occurring hindlimb movements normally obtained in a few days after SCI were acutely suppressed by clonidine application [58]. These authors propose that the suppressive effect of clonidine is mediated rather by the I1- imidazoline than by α2 adrenergic receptors what can be supported by the comparable affinity of clonidine to both α2 and I1 receptors (Ki = 31.62 – 47.00 nM vs. Ki = 31.6228 nM; https://pdsp.unc.edu/databases/kidb.php). Also in the yohimbine effect, the other set of receptors (i.e. 5-HT1A, D2) might be involved when considering rather significant yohimbine affinity to them (Ki = 125.89 nM vs. Ki = 339 nM respectively). Obviously, the role of these types of receptors in conducting the clonidine or yohimbine modulation of locomotor performances might be considered but it is beyond the scope of our investigation and needs further clarification..”

 Results

section 2.2. Some quantification of the anatomical data described would greatly strengthen this section.

We would like to thank the Reviewer for this suggestion and according to it we have added morphological analysis in the two chapters of the Results sections:

2.1. 5-HT and NA Neurons in the Donor Brainstem

In this section, we have included a description of the embryonic region taken for grafting and we specified that each dissection contains typically 110,000-120,000 cells of which approximately 3% are serotonin (5-HT) neurons medially located in the B1/B2/B3 groups (first paragraph – lines 71-78).   

And in

2.2. 5-HT and NA Neurons in the Grafted Spinal Cord

Now, we have described data of detailed examination performed on serial sections, in which we established the numbers of 5-HT and NA neurons that survived in the graft area in our experimental rats (second paragraph – lines 115-122).

“Detailed examination of serial sections immunostaining revealed that 5-HT/TPH neurons preferentially occupy areas in close proximity to the rim of the grafted tissue. We quantified serotonergic neurons and found from 106 to 464 5-HT/TPH positive cells per graft with the length ranged between 720 to 1440 mm. On average, we identified 270 ± 143 5-HT/TPH cells per graft, which means that the viability of the fetal cells at 3 months after transplantation accounts for around 10%. However, it needs to be noticed, that both total cell number and their viability was underestimated since cells were counted in every fifth section of 12 mm thickness, each 70 µm apart (n = 5 grafted rats).

And lines 126-129

“Within the mentioned area of grafted spinal cords, we identified from 7 to 164 NA (TH positive) neurons, on average 87 ± 67 cells per graft (Figure 2 J, notice that the total number of NA cells was underestimated as in case of 5-HT/TPH-positive cells).”

 line151: “Contrary to our expectation, …”. Needs explanation.

We have removed this controversial statement (line 183).

 Figures

Individual data points (from the 6 grafted and 5 not-grafted rats) should be added to all bar graphs.

The individual data points have been added to all bar graphs.

Figures 4 and 5: For consistency with other figs. and to help better understand F stats reports, pre-drug conditions should be labeled “pre-Yoh or pre-Clon”.

The labels “pre-Yoh or pre-Clon” have been incorporated to Figure 4 (now Fig. 5) and Figure 5 (now Fig. 6)

I suggest merging Figs.7 and 8 into one single figure and Figs.9 and 10 into another.

Figure 7 and Figure 8 have been merged into one FIGURE 7, as well as Figure 9 and Figure 10 merged into one FIGURE 8

 Discussion

line 424: “…fully differentiated…” requires clarification.

Our intention was to underline that the 5HT cells harvested for transplantation at E14 were postmitotic and differentiated since they did not accumulate EdU already at E13 and expressed serotonin at E14 as we showed in the original submission. Therefore, we have now modified two respective sentences, as follows: 

lines 134-137: “Using EdU (5-Ethynyl-2-deoxyuridine) injected to the pregnant rat at E13 we found that 5-HT and NA neurons were already postmitotic at E14 when the brainstem tissue was harvested for grafting since we did not observe accumulation of EdU in cells that gave rise to neurons at E14 (Figure 3 A, B, J).

and

lines 414-416:  “In addition, we found that the 5-HT and NA neurons were postmitotic and differentiated at the time of embryonic tissue dissection for grafting and that the host environment did not stimulate their proliferation and differentiation.”

lines 449-456: The postsynaptic scenario is difficult to follow.

The postsynaptic scenario, as well as the other parts of the discussion, have been modified in the aim to clarify our statements.

line 461-462: after spinal cord injury? Also in ref 39 refers to clonidine induced depression of descending-evoked responses in commissural interneurons in intact cord.

Now, we have taken care of correcting all the information regarding if they were established in normal or injured condition. We have also made clear any similarities or differences between rats, cats or mice in the literature data discussed here.  

line 512-515: could sex differences be included as well?

We have included sex differences in our final conclusion (lines 521 – 524) and now the sentence sounds as follows: “These results show clearly that the effects of noradrenergic pharmacotherapy (such as clonidine or yohimbine application) may depend on whether or not the spinal lesion is complete, whether it is acute or chronic state, as well as sex differences and the interspecies differences, have to be considered.

 Methods

line 644: RM not defined

The definition of RM has been added in the Methods section: 4.8. Statistical analysis (line 664).

Conclusions

lines 650-651: suggest synaptic connection. These may or may not be functional.

This aspect of suggested synaptic connection is now clarified (Lines 669-671) as follows:

“As we described above our morphological data show the 5-HT neurons surrounded by NA axons as well as the NA neurons surrounded by 5-HT axons in the graft region suggesting functional synaptic connections between them or at least volume transmission might be considered.”

lines 676-678: reciprocal connections are not supported by results shown.

We agree with the Reviewer that reciprocal connections are nor supported by our results. We took care to emphasize in the Discussion and Conclusion sections that reciprocal interaction “might be essential”. In these chapters, we discussed all the possible mechanisms which could be modulated by restored 5-HT/NA innervation by our grafting strategy. And  we conclude that “Whether the mechanisms that are responsible for locomotor recovery after intraspinal grafting remain those from the intact motor control system needs further investigations.” (Lines 721-723).

Reviewer 2 Report

This paper describes the recovery of hindlimb locomotor activity induced by intraspinal grafting of embryonic brainstem tissue containing 5HT and NA neurons after spinal cord transection. Based on their experimental findings following treatment with an α2 adrenergic receptor antagonist and agonist, the authors speculate that grafted NA neurons play a role in locomotor recovery. This paper contains some interesting content and the reciprocal innervation of grafted 5HT and NA neurons is especially worth further consideration.

Major comments

1 The authors described that the most caudal NA fibers were observed around the central canal, 7 mm below the graft. This level is estimated to be the caudal-most part of the thoracic cord in 5-6 month old rats. Most of motoneurons of the soleus and tibialis anterior are located in the ventral horns of L3 and L4. Most of the CPG neurons involved in hindlimb locomotion are located in the lumbar region. Therefore, it seems unlikely that grafted NA neurons would innervate these neurons directly.

2 The speculation that the grafted NA neurons directly influence the CPG/motoneurons via α2 adrenergic receptors is not supported by the present morphological experiment. Therefore, the authors should reconsider their conclusion that the direct effect of the NA-descending system on the CPG/motoneurons was achieved via the intraspinally grafted neurons. Accordingly, a significant modification of Figure 11 is also needed.

Specific comment

1 Innervation of the intermediolateral zone by NA fibers is not shown in Fig. 2, but should be shown.

2 The authors described that 5HT fibers were found as much as 20 mm from the graft caudally. Innervation of the lumbar spinal segments by 5HT fibers should be shown in Fig. 2.   

Author Response

This paper describes the recovery of hindlimb locomotor activity induced by intraspinal grafting of embryonic brainstem tissue containing 5HT and NA neurons after spinal cord transection. Based on their experimental findings following treatment with an α2 adrenergic receptor antagonist and agonist, the authors speculate that grafted NA neurons play a role in locomotor recovery. This paper contains some interesting content and the reciprocal innervation of grafted 5HT and NA neurons is especially worth further consideration.

We are very pleased with the effort taken by the Reviewer. We corrected our manuscript according to the Reviewer’s suggestions. We hope our corrections improved the text of our manuscript and made it clear now.

Major comments

1 The authors described that the most caudal NA fibers were observed around the central canal, 7 mm below the graft. This level is estimated to be the caudal-most part of the thoracic cord in 5-6 month old rats. Most of motoneurons of the soleus and tibialis anterior are located in the ventral horns of L3 and L4. Most of the CPG neurons involved in hindlimb locomotion are located in the lumbar region. Therefore, it seems unlikely that grafted NA neurons would innervate these neurons directly.

2 The speculation is not supported by the present morphological experiment. Therefore, the authors should reconsider their conclusion that the direct effect of the NA-descending system on the CPG/motoneurons was achieved via the intraspinally grafted neurons. Accordingly, a significant modification of Figure 11 is also needed.

Thank you for focusing our attention on this issue that needs some additional explanation. Here it is an answer to both comments.

In the revised manuscript we have now provided additional information in Figure 10 showing anatomical details of WAG spinal cord. Please notice that this rat strain is much smaller than Wistar and the spinal cord is adequately smaller and the segments are shorter.

We have expanded examples of immunohistochemical data in Figure 2 (now it is Fig.3) and incorporated the information about identified spinal cord segmental levels in which NA and 5-HT fibers were observed. We have also improved the description of our findings. As we show in the corrected figure, the NA fibers were observed 7mm caudally to the graft epicenter which was in the L3 lumbar segment. The majority of NA fibers were present around the central canal but also some fibers were infiltrating the MN area in the ventral horns. The presence of NA fibers in the L3 lumbar segment allowed us to conclude that the grafted NA neurons could directly influence the CPG network via α2 adrenergic receptors (red field arrow in the diagram). We agree that our results did not support the notion of direct action on MNs, and this is why the red arrow in the diagram is dashed and empty, in contrast to that field one indicating the possibility of direct action on CPG. Indirect action of NA innervation on MNs is then suggested.

Specific comment

1 Innervation of the intermediolateral zone by NA fibers is not shown in Fig. , but should be shown.

We have modified the Figure 3 (old Fig 2) by adding an example of the IML zone with NA innervation (L). We have also included an example of occasional NA fiber detected in the MN in the ventral horn (O) in addition to robust innervation detected in the central canal area of the L3 lumbar segment (N). We have also included examples of 5-HT and NA innervation present in the IML in Figure 4 E.

2 The authors described that 5HT fibers were found as much as 20 mm from the graft caudally. Innervation of the lumbar spinal segments by 5HT fibers should be shown in Fig. 2.   

We have modified the Figure 3 (old figure 2) by adding an example of 5-HT innervation of the S1 spinal cord level (G, H, I) that corresponds to the distance about 20 mm from the graft epicenter.

Reviewer 3 Report

The manuscript, entitled “Noradrenergic components of locomotor recovery induced by intraspinal grafting of the embryonic brainstem in adult paraplegic rats”, described experiments about transplantation of embryonic noradrenergic neurons and improvement of motor function. Although authors reported functional recovery, it lacked strong histological evidence to support the conclusion. This reviewer has some major concerns.

  1. In the methods, a lot of basic information was missing, such as the size of tissue block for transplant. How many neurons or cells did the transplant contain? What was the location of the transplantation, the gray matter or white matter, the dorsal or ventral part? 2 ul is not suitable to measure a solid piece. Histological results showed some grafted and survived 5-HT+ and TH+ neurons. However, there was no evidence for whole grafted tissue survival and integration.
  2. There was no evidence about grafted-derived noradrenergic axons project to MNs in the L/S spinal cord.
  3. In the result, it needs to describe main findings and use representative figures as the evidence. However, this manuscript seems to do it oppositely. The result part mainly contains the description of the figures.
  4. In the discussion, authors mentioned several species, including rats, cats and mice. This caused messing and complicated situation to prevent passing clear information.
  5. Authors talked about different responses in naïve rats, those with full transection, or incomplete SCI for two drugs yohimbine and clonidine. However, the explanation was not convincing.

Author Response

The manuscript, entitled “Noradrenergic components of locomotor recovery induced by intraspinal grafting of the embryonic brainstem in adult paraplegic rats”, described experiments about transplantation of embryonic noradrenergic neurons and improvement of motor function. Although authors reported functional recovery, it lacked strong histological evidence to support the conclusion. This reviewer has some major concerns.

  1. In the methods, a lot of basic information was missing, such as the size of tissue block for transplant. How many neurons or cells did the transplant contain? What was the location of the transplantation, the gray matter or white matter, the dorsal or ventral part? 2 ul is not suitable to measure a solid piece. Histological results showed some grafted and survived 5-HT+ and TH+ neurons. However, there was no evidence for whole grafted tissue survival and integration.

We apologize for not having addressed these very valid points more clearly in our original submission. In our revised manuscript we have provided all the missing information including quantification of survived 5-HT+ and TH+ neurons. The required amendments have been made as follows:

Lines 88 - 97:

"First, we investigated the graft tissue survival and its possible integration with the host spinal cord environment. In all spinal transplanted rats, the presence of grafted tissue was positively verified. The grafted tissue was detected predominantly in the grey matter of the spinal cord as aggregates with about 1000-1500 µm in diameter surrounded by host tissue (Figure 2). Detection of the robust serotonergic fibers infiltrating the host spinal cord areas outside the graft region (i.e. the ventral horns, Figure 2 A) indicates successful integration of the grafted tissue within the host spinal cord environment. On the other hand, GFAP (glial fibrillary acidic protein) staining demonstrated the absence of reactive astrocytes on the border of the grafted tissue and the host spinal cord (Figure 2 B). These results indicate that embryonic tissue easily integrates within the host spinal cord and does not evoke long-lasting astrogliosis that, if present, might have created a solid scar."

Lines 553 - 559:

"The embryonic raphe B1-B3 area, approximately 0.25 x 0.5 x 1 mm, was dissected under a surgical microscope. For grafting procedure, a sharpened glass micropipette connected to a Hamilton syringe was used. A dissected solid piece of embryonic tissue was injected into a host rat spinal cord 1 mm below the pial surface through a small laminectomy performed at the T10/11 vertebrae level of the host spinal cord, which corresponds to the Th12 thoracic spinal cord level (Figure 10)."

  1. There was no evidence about grafted-derived noradrenergic axons project to MNs in the L/S spinal cord.

We would like to thank the Reviewer for focusing our attention on this issue that needs some additional explanation.

In the revised manuscript we have now provided additional information in Figure 10 showing anatomical details of WAG spinal cord. Please notice that the WAG rat strain is much smaller than Wistar and the spinal cord is adequately smaller and the segments are shorter.

We have expanded examples of immunohistochemical data in Figure 2 (now it is Fig. 3) and incorporated the information about identified spinal cord segmental levels in which NA and 5-HT fibers were observed. We have also improved the description of our findings. As we show in the corrected figure, the NA fibers were observed 7mm caudally to the graft epicenter which was in the L3 lumbar segment. The majority of NA fibers were present around the central canal but also some fibers were infiltrating the MN area in the ventral horns. The presence of NA fibers in the L3 lumbar segment allowed us to conclude that the grafted NA neurons could directly influence the CPG network via α2 adrenergic receptors (red field arrow in the diagram). We agree that our results did not support the notion of direct action on MNs, and this is why we used in our schematic diagram (Figure 11) dashed lines to indicate the normal connections, not reestablished in our study. In contrast, the solid lines with field arrows indicate the possibility of direct action on CPG. Indirect action of NA innervation on MNs is then suggested.

  1. In the result, it needs to describe main findings and use representative figures as the evidence. However, this manuscript seems to do it oppositely. The result part mainly contains the description of the figures.

The Results part of the manuscript has been improved as requested.

  1. In the discussion, authors mentioned several species, including rats, cats and mice. This caused messing and complicated situation to prevent passing clear information.

The discussion has been improved as requested and particular care was taken to clarify which data come from rats and cats and why it has been mentioned. We have taken care of correcting all the information regarding if they were established in normal or injured condition.

  1. Authors talked about different responses in naïve rats, those with full transection, or incomplete SCI for two drugs yohimbine and clonidine. However, the explanation was not convincing.

We have also made clear any similarities or differences between rats, cats, or mice in the literature data regarding the effects of yohimbine and clonidine applications discussed here. We hope that our corrections improved the text of our manuscript.

Round 2

Reviewer 2 Report

The revised manuscript has been significantly improved. The authors addressed to the queries adequately. However, there are some points that need improvement.

Comments

  1. Page 4, line 110: Should "at the age of"  be "at the edge of".
  2. Page 4, line 111-113: Abbreviations are not needed here.
  3. page 4, line 139: Should Figure 2 J-O be Figure 3....
  4. Page 4, line 141: Should Figure 2 M-O be Figure 3....
  5. Page 7, line 159-163: The legends in Figure 3 do not match the photo panels. Figure 3L-O are missing.

Author Response

The revised manuscript has been significantly improved. The authors addressed to the queries adequately. However, there are some points that need improvement.

We would like to thanks the Reviewer for pointing out our mistakes. We have taken care of correcting all the mentioned errors.

Comments

  1. Page 4, line 110: Should "at the age of"  be "at the edge of".

Line 105: The word “age” has been replaced by “edge”.

  1. Page 4, line 111-113: Abbreviations are not needed here.

The Abbreviations are part of each figure legend, so we would prefer to keep them in Figure 2 as well.  

  1. page 4, line 139: Should Figure 2 J-O be Figure 3....

line 131: Corrected.

  1. Page 4, line 141: Should Figure 2 M-O be Figure 3....

Line 133: Corrected.

  1. Page 7, line 159-163: The legends in Figure 3 do not match the photo panels. Figure 3L-O are missing.

Line 140-142: Figure legend (Fig 3) has been corrected.